# Unlocking Nature’s Rhythms: Insights into Secondary Metabolite Modulation by the Circadian Clock

**DOI:** 10.3390/ijms25137308

**Published:** 2024-07-03

**Authors:** Marina Pérez-Llorca, Maren Müller

**Affiliations:** 1Department of Biology, Health and the Environment, Faculty of Pharmacy and Food Sciences, University of Barcelona, 08028 Barcelona, Spain; 2Institute of Nutrition and Food Safety (INSA-UB), University of Barcelona, 08028 Barcelona, Spain; 3Department of Evolutionary Biology, Ecology and Environmental Sciences, Faculty of Biology, University of Barcelona, 08028 Barcelona, Spain

**Keywords:** carotenoids, circadian clock, flavonoids, phenolic compounds, terpenoids, secondary metabolites

## Abstract

Plants, like many other living organisms, have an internal timekeeper, the circadian clock, which allows them to anticipate photoperiod rhythms and environmental stimuli to optimally adjust plant growth, development, and fitness. These fine-tuned processes depend on the interaction between environmental signals and the internal interactive metabolic network regulated by the circadian clock. Although primary metabolites have received significant attention, the impact of the circadian clock on secondary metabolites remains less explored. Transcriptome analyses revealed that many genes involved in secondary metabolite biosynthesis exhibit diurnal expression patterns, potentially enhancing stress tolerance. Understanding the interaction mechanisms between the circadian clock and secondary metabolites, including plant defense mechanisms against stress, may facilitate the development of stress-resilient crops and enhance targeted management practices that integrate circadian agricultural strategies, particularly in the face of climate change. In this review, we will delve into the molecular mechanisms underlying circadian rhythms of phenolic compounds, terpenoids, and N-containing compounds.

## 1. Introduction

Plants, like many other living organisms, have evolved an internal oscillator known as the circadian clock. This internal clock allows them to anticipate daily and seasonal environmental variations by generating endogenous rhythms of physiological responses, including changes in transcriptional and post-translational regulation [1,2].

Environmental time cues, termed as “Zeitgebers” by the physician and biologist Jürgen Aschoff [3], synchronize the endogenous timing system to a period of 24 h (i.e., the Earth’s rotation period). The most important Zeitgebers are light/dark cycles and temperature cycles, which establish specific circadian rhythms that regulate physiological processes. Zeitgebers serve as one of the three components of the internal oscillator, functioning as input signals that provide information to the central oscillator. The central oscillator, the second component, includes transcriptional and translational feedback loops. This central oscillator, in turn, drives downstream processes that govern various physiological and metabolic pathways in the plant, known as the output pathway and constituting the third component [4,5,6]. The input pathway is mediated by the red/far-red light-sensing photoreceptors phytochromes A and B (phyA and phyB) and the blue light-sensing cryptochromes (cry1 and cry2) [7]. Phytochromes act additively in red light input to the clock, whereas cryptochromes act redundantly in blue light input [8]. Moreover, cry2 and phyB physically interact in the nucleus, and their mRNA abundances coincide under constant light conditions, suggesting a regulatory loop between cryptochromes and phytochromes for the light input pathway [9,10].

The central oscillator is composed of “canonical clock genes” that work at various genomic levels that compose the central or core loop, the morning loop, and the evening loop [11,12] (Figure 1). The central loop consists of the MYB-type transcription factors CIRCADIAN CLOCK ASSOCIATED 1 (CCA1) and the LATE ELONGATED HYPOCOTYL (LHY) that peak around dawn [13]. CCA 1 and LHY activate the sequential expression of *PSEUDO-RESPONSE REGULATORs* (*PRRs*) *9, 7,* and *5* in the morning. *PRR1*, also known as *Timing of CAB expression 1* (*TOC1*), peaks last of all, near dusk. CCA1 and LHY, together with TOC1, form a negative feedback loop in which CCA1 and LHY inhibit TOC1 in the morning, and TOC 1 inactivates CCA1 and LHY in the evening [5,14]. The PRRs also inhibit CCA1 and LHY until near the next dawn [15]. At the same time, CCA1 and LHY are inhibiting the Evening Complex (EC), which is formed by the proteins EARLY FLOWERING 3 (ELF3) and ELF4, and LUX ARRHYTHMO (LUX) and is a repressor of *PRRs* at night, particularly of *PRR9* [16,17]. The *PRRs*, when their expression fades in the late evening due to TOC1 inhibiting them [14,18], will induce the EC. Falling levels of CCA1 and LHY also allow the expression of the EC. Also, at this time, CCA1 HIKING EXPEDITION (CHE), will be repressing *CCA1* [19]. The F-box protein ZEITLUPE (ZTL) inactivates TOC1 through regulation that is aided by the co-chaperone GIGANTEA (GI), which prevents ZTL and TOC1 interaction during the day and the expression levels of which reach a peak in the middle of the day [20,21,22,23]. TOC1 levels are also stabilized by PRR3 in the evening as PRR3 prevents TOC1 sequestration by ZTL [24,25]. Other important components of the central oscillator, which are not included in Figure 1 to preserve simplicity, include the MYB-like transcription factors RVE8 (Reveille 8), the LUX homolog NOX (also known as BROTHER OF LUX ARRHYTHMO), LWD1 (LIGHT-REGULATED WD1), and LNK1 (NIGHT LIGHT-INDUCIBLE AND CLOCK-REGULATED1). Although RVE8 has its peak expression at dawn and forms an additional feedback loop with PRR5 [15], LWD1 activates the transcription of *PRR9*, regulating period length and photoperiodic flowering [26,27]. LNK1 is another activator that interacts with RVE8, enabling the expression of *PRR5* and *TOC1* [28]. Finally, NOX forms a feedback loop with *CCA1*, making NOX a critical component of the circadian clock [29].

At least 30% of transcripts have been reported to be controlled by the circadian clock in Arabidopsis under constant conditions, but this increases to up to 90% when they are exposed to light and/or temperature [30,31], which is evidence of the adjustments that the oscillator constantly makes to adapt to environmental conditions. Plants that have functional oscillators but that cannot entrain well to the environment will struggle to adapt, leading to poor performance. The circadian oscillator is dynamically plastic, and it adjusts phase and period to abiotic and biotic signals [32]. Not only environmental cues alter the circadian period and phases, but also some other metabolites such as sugars, hormones, or ions can adjust the circadian oscillator [32]. PRR7 and PRR9 have been reported to be key components in this plasticity given their pulse-like expression during the day [33]. In fact, PRR9 and PRR7, along with PRR5, are part of the output pathway of the circadian clock, which involves transcription–translation feedback loops that regulate biological processes from seed germination to environmental stress responses (Figure 1). Particularly, *PRR9* and *PRR7* were found to be critical components of a temperature-sensitive circadian system [34]. A few years later, these findings were confirmed, wherein a triple mutant of *PRR9, PRR7,* and *PRR5* presented a higher expression of the dehydration-responsive element-binding protein 1 (DREB1)/C-repeat binding factors (CBFs), suggesting that these PRRs regulate cold stress responses by inhibiting DREB1/CBFs [35]. Moreover, both PRR5 and PRR7 have been recently reported to positively regulate abscisic acid (ABA) signaling through association with the transcription factor ABSCISIC ACID-INSENSITIVE5 (ABI5) during seed germination [36]. CCA1 and LHY also seem to regulate cold stress responses, but instead of being repressors, they induce the expression of CBFs [37,38]. The EC, in turn, regulates the expression of some *PHYTOCHROME INTERACTING FACTORS* (*PIFs*), which have a direct role in thermomorphogenesis [39,40]. *CCA1* and the EC also have a crucial role in environmental stress responses through the regulation of reactive oxygen species (ROS) homeostasis and signaling [41].

Secondary metabolites play a crucial role in the plant stress response and are partially regulated by the circadian clock. Comprehensive reviews on the interaction between the circadian clock and the primary metabolism, especially focusing on sugars and photosynthesis, already exist (e.g., [42,43,44]). Some have addressed the molecular basis of plant stress responses with a focus on circadian-regulated genes [45,46,47]; however, to our knowledge, none have directly linked secondary metabolite accumulation with circadian clock components, except for [48], which focused on phytochemical variations influenced by environmental factors and noted whether these variations were circadian. In this review, we explore the molecular mechanisms underlying the circadian rhythms of secondary metabolites, such as phenolic compounds, terpenoids, and nitrogen-containing compounds. Given that secondary metabolites typically accumulate under stress conditions, we will inevitably consider both abiotic and biotic cues in our discussion.

## 2. Regulation of Secondary Metabolites by the Circadian Clock

Secondary metabolites are highly reactive bioactive compounds that play multifunctional roles in both defense and environmental interactions. These roles include acting as antioxidants, osmoregulators, allelopathic effectors, antifeeding agents, among others. [49,50]. They usually accumulate under stress conditions, which can have detrimental effects on plant growth, including reductions in leaf number, leaf area, plant height, and productivity [50,51]. In addition, they are responsible for plant colors, aromas, and flavors [52]. Although secondary metabolites are often referred to as organic compounds with no essential roles in maintaining plant life processes, recent studies have shown that the boundaries between primary and secondary metabolism are blurred. Instead, it appears that primary and secondary metabolism form an interactive network, with plant hormones acting as regulatory intermediaries, as reviewed in [50]. Secondary metabolites are usually classified into (i) phenolic compounds (i.e., flavonoids, phenolic acids, lignin, lignans, coumarins, stilbenes, and tannins), (ii) terpenoids (i.e., plant volatiles, sterols, carotenoids, saponins, and glycosides), and (iii) nitrogen-sulfur-containing compounds (i.e., alkaloids, glucosinolates, and cyanogenic glycosides) [49].

Plants synthesize more than 100,000 secondary metabolites through different metabolic pathways, with quantity and quality varying depending on environmental conditions, including pathogen infections, herbivore attacks, salinity, cold, heat, and drought stress [53]. In brief, phenolic compounds are synthesized via the shikimic acid and the malonic acid pathway, where phenylalanine ammonia lyase (PAL) and chalcone synthase (CHS) act as key enzymes, as reviewed in [54]. The transcription factor MYB111 has been reported to regulate flavonoid biosynthesis by binding to specific *cis*-elements in the promoter of chalcone synthase (CHS), flavanone carboxylase (F3H), and flavonol synthase 1 (FLS1) under salt stress in Arabidopsis [55]. Terpenoids are synthesized via the mevalonic acid (MVA) and the 2-*C*-methyl-D-erythriol-4-phosphate (MEP) pathway in the cytosol and plastid, respectively, as reviewed in [56,57]. Isopentenyl pyrophosphate (IPP) and dimethylallyl pyrophosphate (DMAPP), which act as universal precursors for all terpenoids, are derived from pyruvate and glyceraldehyde-3-phosphate. A recent study showed that MYB transcription factors triggered β-carotene biosynthesis by promoting the expression of genes encoding carotenoid isomerase (*AtCRTISO*) and lycopene β-cyclase (*AtLCYB*) leading to increased drought tolerance in transgenic Arabidopsis [58]. Nitrogen-sulfur-containing secondary metabolites are characterized by the presence of nitrogen and sulfur molecules in their structure, as reviewed in [59,60]. Sulfur-containing secondary metabolites represent a relatively small group, including glucosinolates and their degradation products such as thiocyanates, isothiocyanates, epithionitriles, and oxazolidinethiones. Amino acids such as lysine, tyrosine, and tryptophan act as precursors in the biosynthesis of nitrogen-containing secondary metabolites. Several studies reported that alkaloid synthesis increased through the activity of biosynthesis enzymes such as tryptophan decarboxylase and hyoscyamine 6β-hydroxylase under UV-B and salt stress, with WRKY transcription factors being key regulators of the alkaloid metabolism [61,62].

In contrast to primary metabolites, the role of the circadian clock in controlling secondary metabolites has been less studied. Transcriptome analyses have revealed that numerous genes involved in the biosynthesis of secondary metabolites exhibit diurnal expression patterns, potentially contributing to enhanced stress tolerance [1]. Most abiotic and biotic stressors, such as pathogen infections, herbivore attacks, cold, salinity, heat, and drought, are associated with increased levels of ROS. The accumulation of ROS in the cell causes oxidative stress, which leads to membrane oxidation, DNA damage, and ultimately cell death [63,64]. However, ROS also act as signaling molecules that activate signal transduction pathways in response to environmental stresses that can compromise plant survival [65,66]. The transcription of ROS-related genes has been shown to be regulated by the circadian clock. Conversely, ROS act as input signals that influence the clock’s transcriptional output [67]. This dual effect of ROS depends on the location of their production, their ability to cross biological membranes, their levels, and their capacity for antioxidant scavenging [68,69]. Therefore, ROS homeostasis is tightly controlled by antioxidant systems, in which secondary metabolites play a crucial role [70,71,72]. Here we discuss new insights into the modulation of secondary metabolites by the circadian clock, which strongly influences growth and stress tolerance responses to biotic and abiotic stresses.

### 2.1. Internal Clock and Phenolic Compounds

Phenolic compounds are among the most common plant compounds in the group of secondary metabolites. Due to their chemical diversity and biological activity, they are important players against abiotic and biotic stress. Phenolic compounds are often antioxidants with protective roles in plants and other organisms involving neutralizing harmful free radicals, which also makes them very interesting and useful as medicines and nutritional supplements for human health. The accumulation and biosynthesis of phenolic compounds (Figure 2) in plants depends on physiological–biochemical, molecular–genetic and environmental factors [73,74]. For instance, various *Brassica* cultivars showed species-specific circadian rhythms in polyphenol content, which correlated with their high antioxidant activity [75].

#### 2.1.1. Phenylpropanoids

Phenylpropanoids are widespread in the plant kingdom and play important roles in plant development and against environmental stress. For instance, they contribute to cell wall structure and act as protective compounds against strong light and UV radiation. In a transcriptomic analysis, numerous genes encoding enzymes in the phenylpropanoid biosynthesis pathway were found to be under the control of the circadian clock and peaked before dawn [12]. The authors of this study suggested that this timing may enable *Arabidopsis* to produce phenylpropanoid-related secondary metabolites to protect cells from sunlight. Furthermore, *Arabidopsis* plants lacking the phenylpropanoid pathway showed increased susceptibility to UV radiation [76,77]. Moreover, plants synthesize a variety of volatile phenylpropanoids and benzenoids (VPBs), which derive from the shikimate pathway via L-phenylalanine [78] (Figure 2). VPBs have signaling functions, particularly for attracting pollinators. However, the emission of VPBs from flowers is restricted to a specific time of day and, for example, *Petunia hybrida* has been characterized as a nocturnal emitter of VPBs [79]. A study by Cheng and colleagues [80] reported that genes of the upstream biosynthesis pathways of VPBs, including *3-deoxy-D-arabinoheptulosonate 7-phosphate synthase* (*DAHPS*), *arogenate dehydratase* (*ADT*), *PAL, cinnamic 4-hydroxylase* (*C4H*), and *4-coumarate:CoA ligase* (*4CL*) are regulated by the circadian clock in *Petunia hybrida*. Moreover, the circadian clock gene *LHY* (*PhLHY*) has been shown to regulate the diurnal expression patterns of VPBs’ pathway in an antiphasic manner. PhLHY protein, which peaks in the morning, binds to cis-regulatory evening elements, which are present in promoters of *ODORANT1* (*ODO1*), a transcription factor gene, and other VPBs genes, thereby restricting their expression to the evening [81].

#### 2.1.2. Flavonoids

Genes involved in the flavonoid biosynthesis pathway are regulated by the interaction of different families of transcription factors such as MYB, basic helix-loop-helix (bHLH), and DW40 proteins. Different combinations of transcription factors and their interactions determine the spatial and temporal activation of these genes [82,83]. The flavonoid biosynthesis pathway involves several positive regulators, including MYB11, MYB12, MYB111, MYB75/PAP1, and MYB-like Domain (AtMYBD). In contrast, MYB-like 2 (MYBL2) has been characterized as a negative regulator. Furthermore, *AtMYBD* expression has been reported to be controlled by diurnal regulation, and it has been suggested that the transcription factor AtMYBD may regulate anthocyanin biosynthesis in a circadian clock-dependent manner [84,85]. In addition, PAP1 has been proposed to act on the circadian regulation of anthocyanin biosynthesis, but further studies are required to validate this [86]. A study by Pérez-García and colleagues [87] revealed that LNK genes and REVEILLE8/LHY-CCA1-LIKE5 (RVE8/LCL5), two circadian components, fine-tune regulation for the precise anthocyanin metabolic pathway through opposite functions. The MYB-related transcription factor RVE8/LCL5 acts in a similar manner to CCA1 and LHY, through similar rhythmic expression that shows a morning acrophase. On the other hand, RVE8/LCL5 and CCA1 have opposing regulatory functions. Whereas RVE8/LCL5 promotes the acetylation level of histone H3 at the *TOC1* promoter, CCA1 represses *TOC1* expression [88]. Thus, RVE8/LCL5 activates not only anthocyanin genes by directly binding to its promoters, but also *TOC1*. However, the latter is activated later during the day by RVE8/LCL5 in contrast to the earlier activation of anthocyanin gene expression. Furthermore, LNK antagonizes RVE8/LCL5 and represses the expression of anthocyanin-biosynthesis genes. Thus, anthocyanin biosynthesis is controlled around midday by the interaction of RVE8/LCL5–LNK, which have opposing functions of controlling *TOC1* gene expression compared to CCA1 [87]. A recent study found similar results in floral tissues of *Crocus sativus* [89]. The authors observed that a MYB-related RVE8-type transcription factor, CstMYB1R1, regulates the circadian clock-specific flavonoid and anthocyanin-biosynthesis pathway with accumulation peaks at dawn and dusk and minimum contents at night. In addition, the *CstMYB1R1* expression correlated with *ANS* and *LEUCOANTHOCYANIDIN DIOXYGENASE* (*LDOX)* gene expression, which work in parallel pathways to generate different proanthocyanins. In addition, in transformed *Nicotiana* plants *CstMYB1R1* expression resulted in increased flavonoid and anthocyanin contents and enhanced abiotic stress tolerance [89]. In potatoes, the *GIGANTEA* (*GI*) gene, *StGI.04*, has been found to promote anthocyanin biosynthesis in tuber skins [90]. Previous studies have already reported that *StGI.04* influences the circadian clock in other plants [91], suggesting that anthocyanin synthesis may be regulated by *StGI.04* in a circadian clock-like manner. CHS is one of the key enzymes in the biosynthesis of flavonoids, and it catalyzes the first step in flavonoid metabolism between p-coumaroyl-CoA and malonyl-CoA to form naringenin chalcone (Figure 2). More than 20 years ago, it was discovered that in the roots and leaves of Arabidopsis the CHS promoter is controlled by the circadian clock [92]. A recent study revealed that in CHS-deficient Arabidopsis seedlings *CCA1* and *TOC1* expression was altered [93]. These findings were confirmed in mutant lines lacking flavonoid 30-hydroxylase activity (F30H), which were unable to synthesize dihydroxylated B-ring flavonoids. Analysis of the microarray data from a previous study [94] provided evidence that flavonoid accumulation is circadian-regulated and adapts depending on light conditions [93]. The authors further suggested, based on their findings regarding the effect of the WD-repeat protein TRANSPARENT TESTA GLABRA1 (TTG1) on clock regulators WD1 and WD2, that flavonoids themselves may affect the circadian clock. Moreover, considering that flavonoids contribute to the maintenance of ROS homeostasis and that there is a strong connection between the clock and the intracellular redox state [67,93], flavonoids appear to help in protecting the integrity of the plant clock under stress conditions. Furthermore, increased expression of the *LHY* orthologue *MtLHY* resulted in enhanced salt tolerance by activating the flavonol synthase gene, *MtFLS*, in *Medicago truncatula* [70]. In apple (*Malus domestica*), two R2R3 MYB transcription factors, MYB88 and the paralogous FLP (MYB124), were found to regulate the expression of the *COLD SHOCK DOMAIN PROTEIN 3* (*MdCSP3*) and *MdCCA1* genes. Moreover, these transcription factors also seemed to modulate anthocyanin accumulation, thereby detoxifying ROS under cold stress [95]. In addition, enhanced expression of genes related to the circadian clock pathway, including *PRR5*, *FLOWERING LOCUS T* gene (*FT*), and *LHY*, were observed in flavonoid-hyper-accumulating *Glycyrrhiza glabra* lines [96], suggesting that the connection between flavonoids and the circadian clock is widespread in the plant kingdom. However, further studies with different species are still needed in order to fully understand the mechanisms underlying the interaction of flavonoids and circadian clock.

### 2.2. Internal Clock and Terpenoids

Terpenoids, also known as isoprenoids, which are classified as secondary metabolites, include plant organic volatiles, sterols, carotenoids, and glucosides among other molecules [97]. These groups have diverse structures and functions, but they are all derived from IPP and its allylic isomer DMAPP, which derive from the cytosolic MVA pathway and the plastidial MEP pathway [98,99] (Figure 3). Terpenoids are involved in defense mechanisms against both biotic and abiotic stresses [56], in photoprotection [100] and in plant reproduction and plant–plant interactions [101,102]. Given that many of the metabolites of the MEP and MVA pathways are products of photosynthesis or are involved in it, it is not surprising that MVA and MEP genes are circadian-regulated [103,104]. Generally, in shoots, genes of the MEP pathway such as *1-deoxy-D-xylulose 5-phosphate* (*DXP*) *synthase* (*DXS*) and *1-hydroxy-2-methyl-2-butenyl 4-diphosphate* (*HMBPP*) *reductase* (*HDR*) are co-expressed with clock genes from the morning loop (i.e., *CCA1*, *LHY* and *PRR9*) whereas MVA pathway genes such as *acetoacetyl-CoA thiolase 2* (*AACT2*) correlate with *TOC1* of the evening loop [105]. Moreover, phyB and PIFs also control the light-dependent response of MVA- and MEP-pathway genes [106,107].

#### 2.2.1. Volatile Terpenes

Volatile terpenes are well-known for being regulated by the circadian clock [108]. As compounds that allow plants to adapt and interact with their surroundings, volatile terpenes need to encompass nature’s rhythms. Volatile terpenes such as β-ionone or β-pinene, which confer plant scent, have been reported to have a diurnal rhythm [109,110]. Although there are studies that clearly link circadian clock components with other types of plant volatiles, e.g., [81,111], there is a gap of knowledge for volatile terpenes. The work of Wilkinson et al. [112] tentatively connected *CCA1/LHY* expression with isoprene emission rhythms in palm oil. A year later, these findings were supported by the study of Loivamäki et al. [113], who found that AtLHY proteins bound to the *ISPS* (*Isoprene Synthase*) promoter fragments of *Populus x canescens.*

#### 2.2.2. Sterols

In addition to being an essential component of cell membranes, plant sterols also play a crucial role in plant growth and development as well as in the response to abiotic stress [114]. Sterols may be regulated by some components of the circadian clock, although direct regulation by canonical clock components still remains elusive. Sterol metabolites were increased in a *cry2* mutant, while carotenoids were reduced, confirming cry2’s crucial role in fruit ripening [115]. Moreover, a recent study reported that sterols might also interact with PIFs to promote hypocotyl growth in Arabidopsis low-red- and far-red-light-treated seedlings [116].

#### 2.2.3. Carotenoids

Carotenoids are essential photosynthetic pigments that play a crucial role in light harvesting and photoprotection. Additionally, they exhibit potent antioxidant activity as they are scavengers of ROS [117]. Carotenoids can be classified into carotenes, with β-carotene as their major representative, and into xanthophylls. Carotenoid biosynthesis is clock-regulated. In a global transcriptome analysis, diverse genes of the carotenoid biosynthesis pathway were found to be clock-controlled [31]. Particularly, genes encoding enzymes that are involved in the synthesis of carotenoids from GGDP were found to be 83% enriched in circadian regulation and peaked at subjective dawn. Indeed, it has been reported that PRRs, whose expression peaks in the morning, are negative regulators of the MEP-pathway’s metabolic routes, resulting in carotenoids being upregulated [118]. PHYTOENE SYNTHASE (PSY), the first enzyme involved in carotenoid biosynthesis [119], was also revealed to be clock-controlled [31]. *PSY* expression is also regulated by PIFs, which act as repressors of carotenoid accumulation during seedling de-etiolation [106]. It was later found that *PSY* expression was mediated by the transcription factor ELONGATED HYPOCOTYL5 (HY5) [120], a discovery showing the antagonistic roles of PIFs and HY5 in carotenoid biosynthesis. Finally, the study of Convington and colleagues also found that the transcripts of a gene encoding violaxanthin de-epoxidase (i.e., *NPQ1*) peaked at subjective dusk [31], which is in keeping with the dissipation of energy as heat by the xanthophyll cycle at the end of the day [121]. Most recently, the MYB-like transcription factor REVEILLE 1 (RVE1), which promotes growth [122], has been reported to positively regulate carotenoid levels in rice [123].

### 2.3. Internal Clock and N-Containing Compounds

Secondary metabolites containing nitrogen (N) in plants typically specialize in plant defense. These metabolites include non-protein amino acids, alkaloids, glucosinolates, indole phytoalexins, and cyanogenic glycosides. N-containing secondary metabolites are synthesized from aromatic and aliphatic amino acids that originate from the shikimate pathway and the Krebs cycle, respectively (Figure 4) [124]. Given that N metabolism is controlled by the circadian clock [125], it could be hypothesized that certain secondary metabolites containing N might be circadian-controlled.

#### 2.3.1. Non-Protein Amino Acids

Non-protein amino acids found as secondary metabolites in plants have antiherbivore antimicrobial and allelopathic activities, among others [126,127]. Although some non-protein amino acids, like gamma-aminobutyric acid (GABA), are ubiquitous across nearly all plant species, others are specific to certain plant families. For instance, legumes are known for being rich in non-protein amino acids such as canavanine or mimosine [128,129]. We found no studies addressing the regulation of non-protein amino acids by the circadian clock, only a few referring to the daily fluctuations of GABA. The accumulation of GABA and the expression of its catalyst, glutamate decarboxylase, appear to follow a coordinated response during the day, particularly under stress conditions [130,131,132]. However, there is no evidence to suggest that this process is influenced by clock-related genes. The lack of evidence of circadian-regulated non-protein amino acids might be attributed to their specific functions in certain plant species. One of their primary roles, as nitrogen-containing compounds, is to serve as nitrogen storage [133]. However, non-protein amino acids that can be incorporated into proteins, especially those homologous to common amino acids in plants like canavanine, may be indirectly regulated by the circadian clock. This is suggested by the fact that some ubiquitin-specific proteases (UBPs), which are known to regulate the circadian rhythm of clock genes, are also required for the degradation of canavanine-containing proteins [134,135]. Nevertheless, further research would be needed to test these hypotheses.

#### 2.3.2. Alkaloids

Alkaloids are N-containing compounds of which there are over 20 different classes [136]. They can be both beneficial and toxic and are used in the pharmaceutical industries due to such properties [137]. Well-studied alkaloids are opium alkaloids such as morphine that are present in Papaveraceae, and Solanum alkaloids such as solanine that are present in *Solanum* spp. [138,139]. There is no direct evidence of alkaloids being regulated by circadian clock genes even though some compounds such as quinolizidine alkaloids do fluctuate during the day [48].

#### 2.3.3. Glucosinolates

Glucosinolates (GSL) are sulfur-rich secondary metabolites, the structure of which contains a side amino acid chain that can be of an aliphatic, aromatic, or indole nature [140]. GSL are mainly found in *Brassica* spp. and they respond to both biotic and abiotic factors [141,142]. It is well-established that GSL levels are linked to pathogens’ feeding habits [143], which has led to the hypothesis that these defenses might have a daily rhythmicity. As a matter of fact, *Brassica* species do keep track of time to coordinate their defenses [75,144]. Higher accumulation of GSL has been reported during the day compared to the night, which is concomitant with daily sulfate rhythms [145]. Moreover, the transcription factor HY5, which acts downstream of multiple photoreceptors [146], has been proven to regulate GLS biosynthesis genes [145,147]. *CCA1* induces indole GSL biosynthesis, most likely through transcriptional activation [148]. More recently, the co-chaperone GI has been found to be involved in GSL biosynthesis, as a GI-knockout mutant revealed altered GSL transcripts, particularly those of aliphatic GSL, and reduced GSL contents in Chinese cabbage [149]. Interestingly, GSL have also been reported to regulate the circadian clock. Arabidopsis mutants with loss of function of *AOP2*, a key gene in the synthesis of aliphatic GSL, exhibited an altered circadian period [150], suggesting that GSL act as feedback regulatory inputs and contribute to the plasticity of the circadian oscillator [151].

#### 2.3.4. Phytoalexins

Indole phytoalexins derive from indole GSL degradation, thus making indole phytoalexins S- and N-containing compounds [152]. Although deciphering phytoalexins regulation is a complex matter given that they are synthesized from remote precursors through the *de novo* synthesis of enzymes [153], a link between the circadian clock and certain phytoalexins has been reported. GI directly binds to the intronic region of *PAD4* (*PHYTOALEXIN DEFFICIENT4*), a key gene in the last steps of the synthesis of camalexin, which is a phytoalexin in Brassicaceae [154]. *CCA1* did not seem to directly regulate camalexin upon aphid infection [148], despite the fact that indole phytoalexins derive from indole GSL and GSL were proven to be regulated by this central clock component, suggesting that there may be indirect regulatory mechanisms in play.

#### 2.3.5. Cyanogenic Glucosides

Cyanogenic glucosides are N-containing bioactive compounds that are well-known for being effective herbivore deterrents [155]. Although cyanogenic glucosides in high concentrations can be lethal to animals, some insects feed preferentially on cyanogenic plants, demonstrating a tight evolutionary link between plants and insects [156,157]. Cyanogenic glucoside biosynthesis has been found to vary during the day, with linamarin, a cyanogenic glucoside of cassava, decreasing with light and the expression of its biosynthetic genes peaking in the morning and decreasing at dusk [158]. This potential regulation by light aligns with the findings of Kongsawadworakul and colleagues [159], who observed that young plants of *Hevea brasiliensis* exposed to direct sunlight in the morning showed decreased contents of cyanogenic glucosides compared to those of plants in the shade. The precise mechanism behind this light regulation of cyanogenic glucosides remains unclear, and further studies using photoreceptor mutants could provide valuable insights.

In summary, numerous secondary metabolites have been found to follow diurnal rhythms, with several being directly regulated by canonical clock genes (Table 1). Although examples have been observed in species beyond Arabidopsis, most of the evidence is based on this model plant species. Evidence from non-model species often reports daily rhythms but lacks direct proof of circadian clock gene regulation of these secondary metabolites. This highlights the need for more genomic and transcriptomic studies in non-model species to better understand these regulatory mechanisms. Additionally, understanding these mechanisms is crucial for both crop breeding and agriculture in a climate change context as altering the production of secondary metabolites or optimizing crop-harvest times (see [160]) could provide plants with enhanced tolerance and resistance to environmental stressors as well as improved nutritional quality.

## 3. Conclusions

Numerous secondary metabolites have been found to follow daily rhythms, such as carotenoids, flavonoids, and glucosinolates, with some of them being directly regulated by canonical clock genes such as *CCA1* and *TOC1* (Table 1). Interestingly, the MYB-related transcription factor RVE8/LCL5 activates the anthocyanin-biosynthesis genes with a similar rhythmic expression as *CCA1*, showing a morning acrophase. Unlike CCA1, RVE8/LCL5 activates *TOC1* expression, whereas LNK acts as its antagonist, inhibiting the anthocyanin-biosynthesis genes. Our findings underscore the pivotal role of the circadian clock in mediating plant responses to both abiotic and biotic environmental challenges. As illustrated in Figure 5, the circadian clock not only regulates the production of secondary metabolites and stress signaling pathways in a spatial and temporal manner, but also integrates feedback loops that modulate its function and contribute to its plasticity. In addition, secondary metabolites appear to play a crucial role in protecting the circadian clock by acting as antioxidants. However, most molecular studies are still based on the model plant Arabidopsis, whereas in non-model species, direct evidence for the regulation of these secondary metabolites by circadian clock genes is lacking. This highlights the need for further genomic and transcriptomic studies in non-model species with cropping or ecological value to better understand these regulatory mechanisms and to implement local and targeted agricultural practices. Additionally, further research is needed to identify secondary metabolites that alter the circadian period and/or phases. Understanding the intricate network of circadian regulation is essential for improving plant resilience and adaptation. In the context of climate change, this knowledge is particularly crucial for crop species. It could lead to more directed crop breeding and more efficient, cost-effective management practices that integrate circadian agriculture as a fundamental aspect. By incorporating these insights into breeding programs and agricultural-management decisions, we can contribute to sustainable agricultural practices and enhance food security.

## Figures and Tables

**Figure 1 ijms-25-07308-f001:**
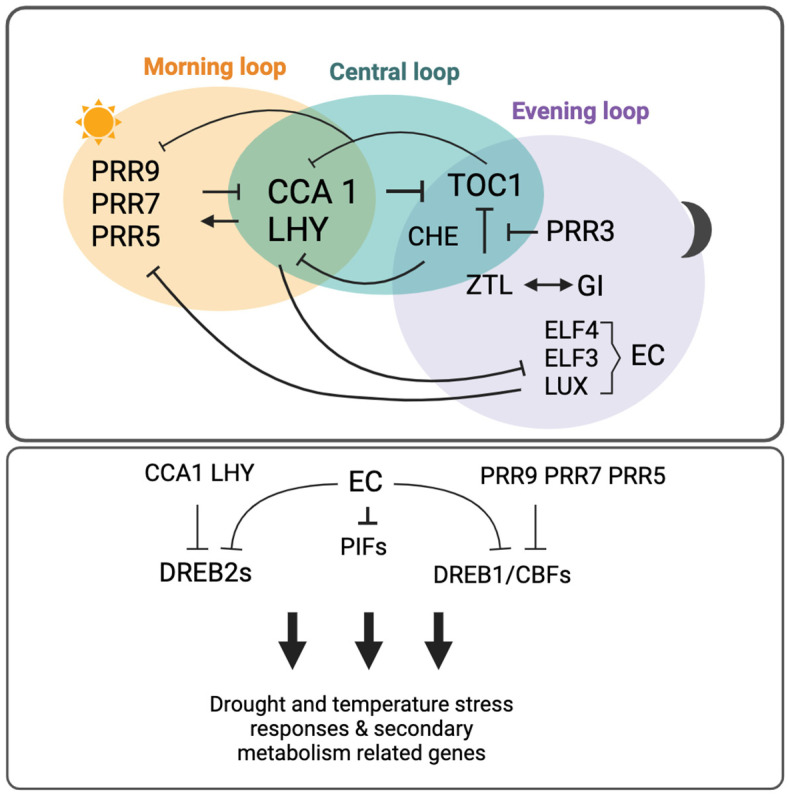
A simplified diagram of the central oscillator and its components, and transcription–translation loops that regulate biological processes. The central oscillator is composed of a central loop, a morning loop, and an evening loop. CCA1 and LHY repress TOC1 (PRR1) in the morning and sequentially activate the other PRRs during the day. TOC1, during the day, is also inactivated by ZTL with the help of GI. CCA1 and LHY inactivate EC during the day. TOC1, the levels of which are stabilized by PRR3 that sequesters ZTL, inactivates CCA1 and LHY as well as PRRs in the evening. Concomitantly, CCA1 is also repressed by CHE. In the evening, EC represses PRRs. The EC represses PRRs at night. As part of the transcriptional–translational loops, CCA1 and LHY repress DREB2s, whereas PRR9, PRR7, and PRR5 repress DREB1/CBFs. The EC also acts as a repressor of DREB2s and DREB1/CBFs as well as PIFs. The repression of these transcriptional–translational loops triggers drought and temperature-stress responses, as well as the regulation of genes related to secondary metabolism. For the sake of simplicity neither genes, expression nor proteins have been differentiated when representing the clock components. CBFs, C-repeat binding factors; CCA1, CIRCADIAN CLOCK ASSOCIATED 1; CHE, CCA1 HIKING EXPEDITION; DREB1, dehydration-responsive element-binding protein 1; EC, Evening Complex; ELF3/4, EARLY FLOWERING; GI, GIGANTEA; LHY, LATE ELONGATED HYPOCOTYL; LUX, LUX ARRHYTHMO; PIFs, PHYTOCHROME INTERACTING FACTORS; PRRs, PSEUDO-RESPONSE REGULATORs; TOC1, Timing of CAB expression 1; ZTL, ZEITLUPE.

**Figure 2 ijms-25-07308-f002:**
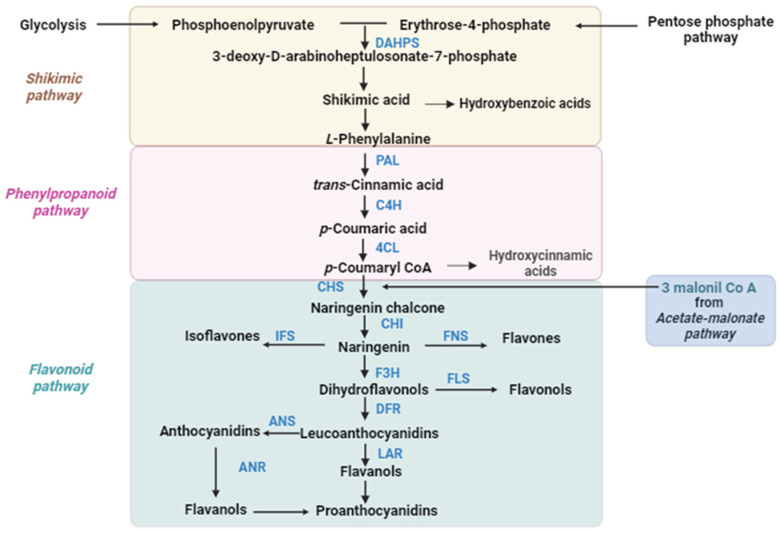
Simplified diagram of phenolic compounds’ biosynthesis pathway. Abbreviations: ANR, anthocyanidin reductase; ANS, anthocyanidin synthase; CHS, chalcone synthase; C4H, cinnamate 4-hydroxylase; CHI, chalcone-flavanone isomerase; 4CL, 4-coumaroyl-coenzyme A ligase; DAHPS, 3-deoxy-D-arabinoheptulosonate 7-phosphate synthase; DFR, dihydroflavonol 4-reductase; F3H, flavanone 3-hydroxylase; FLS, flavonol synthase; FNS, flavone synthase; IFS, isoflavone synthase; LAR, leucoanthocyanidin reductase; PAL, phenylalanine ammonia lyase.

**Figure 3 ijms-25-07308-f003:**
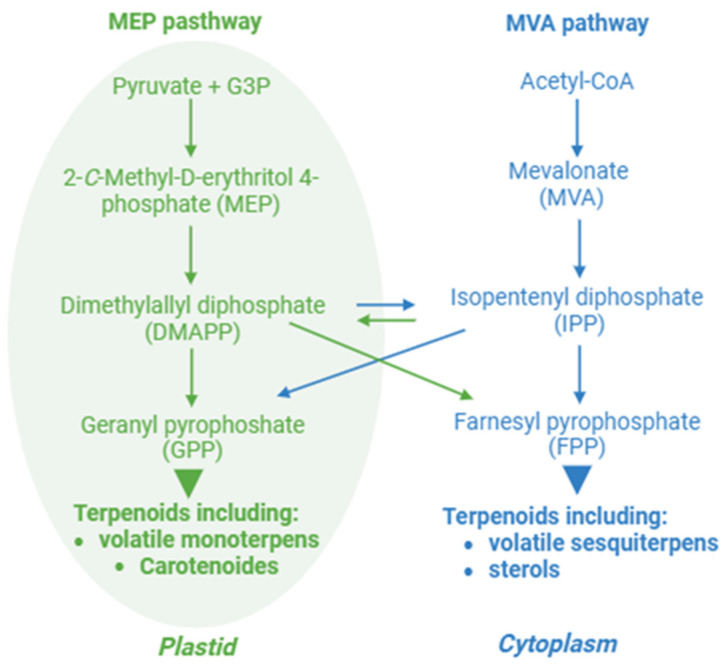
Biosynthesis pathway of terpenoids. The MEP and MVA pathways interact to synthesize terpenoids by providing common precursors (i.e., DMAPP and IPP) and facilitating the transport of these precursors between the plastid and the cytoplasm. G3P, glyceraldehyde-3-phosphate.

**Figure 4 ijms-25-07308-f004:**
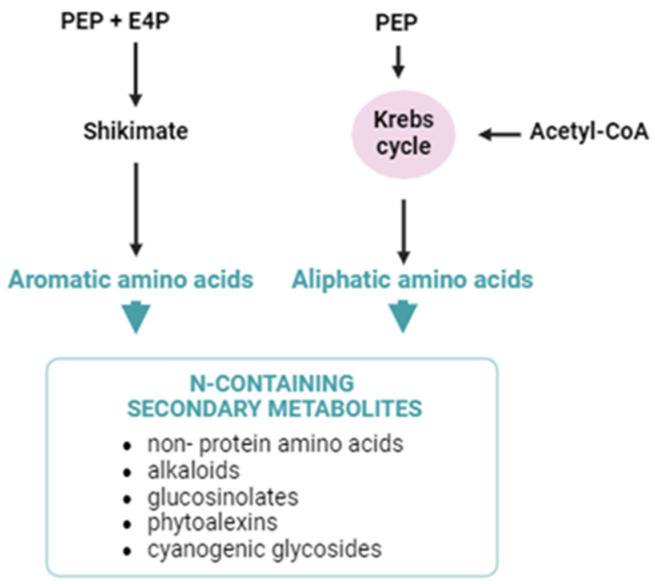
Simplified biosynthesis pathway of N-containing compounds. The shikimate pathway and the Krebs cycle provide aromatic and aliphatic amino acids to form N-containing secondary metabolites. PEP, phosphoenolpyruvate; E4P, erythrose-4-phosphate.

**Figure 5 ijms-25-07308-f005:**
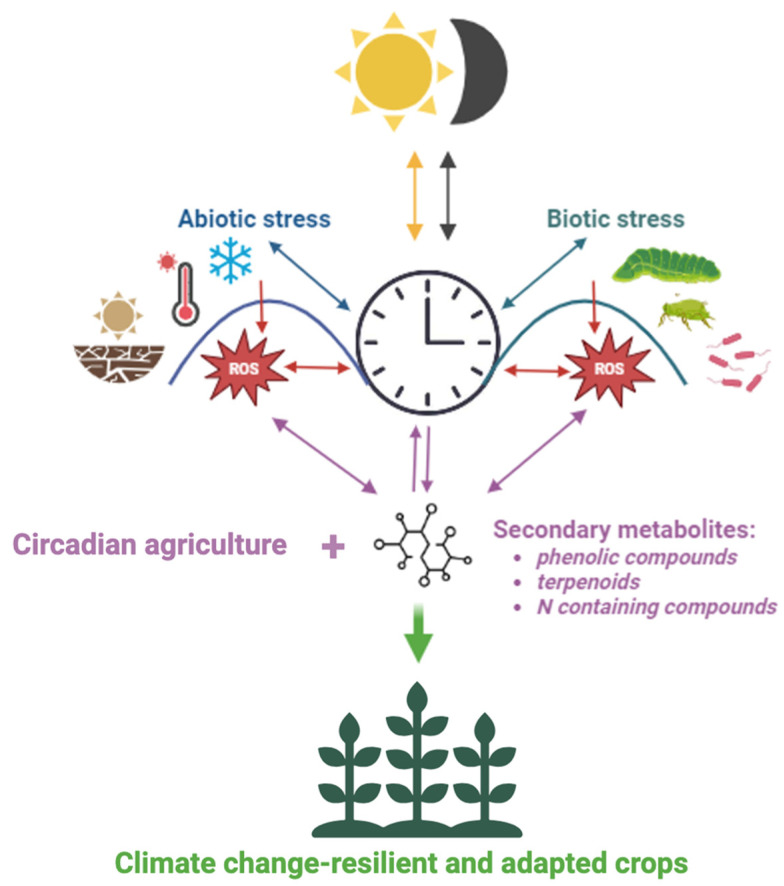
Proposed model indicating the circadian clock as central player in adaptation to abiotic and biotic environmental challenges, with secondary metabolites and stress signaling pathways showing circadian regulation and feedback loops modulating clock function (indicated by bidirectional arrows); based on [49,67,73,93,161]. Incorporating the interaction between circadian regulation and secondary metabolites into plant breeding programs could play a key role in improving the resilience and adaptability of crops to climate change.

**Table 1 ijms-25-07308-t001:** List of the secondary metabolites and secondary metabolite pathways found to be regulated by canonical clock genes.

Plant Species	Clock-Related Genes	Function	Secondary Metabolite	Reference
*Petunia hybrida*	*LHY*	Regulating timing of floral volatile emission by binding to *ODORANT1*	Volatile phenylpropanoid and benzenoid biosynthesis	[81]
*Arabidopsis*	*CCA1, TOC1*	Regulating *CHS* and *F3’H* activity	Flavonoid biosynthesis	[93]
*Glycyrrhiza glabra*	*PRR5, FT, LHY*	Enhanced expression observed in flavonoid-hyperaccumulating lines		[96]
*Medicago truncatula*	*LHY*	Overexpression of *MtLHY* resulted in an increased expression of *MtFLS*, a flavonol synthase gene		[70]
*Crocus sativus*	*CstMYB1R1*	Regulating *ANS* and *LDOX* gene expression resulting in enhanced flavonoid and anthocyanin accumulation with peaks at dawn and dusk and minimum contents at night	Flavonoid and anthocyanin biosynthesis	[89]
*Arabidopsis*	*MYBD*	Thought to act as a regulator of anthocyanin biosynthesis in a circadian-dependent manner	Anthocyanin biosynthesis	[84,85]
	*RVE8/LCL5*	Promoter of anthocyanin-biosynthesis genes		[87]
	*LNK*	Repressor of anthocyanin-biosynthesis genes		[87]
*Arabidopsis*	*CCA1, LHY, PRR9*	Co-expression with *DXS, HDR* from the MEP pathway	MEP pathway	[105]
*Arabidopsis*	*TOC1*	Co-expression with *AACT2* from the MVA pathway	MVA pathway	[105]
*Populus x canescens*	*LHY*	Binding to *ISPS* resulting in peak *ISPS* expression in the morning	Isoprene	[113]
*Arabidopsis*	*PRR 9, PRR7, PRR5*	A triple-knockout mutant showed increased gene expression of carotenoid and ABA biosynthetic pathways	Carotenoid biosynthesis	[118]
*Oryza sativa*	*OsRVE1*	Overexpression of *OsRVE1* increased carotenoid accumulation	Carotenoids	[123]
*Arabidopsis*	*CCA1*	An overexpression line of *CCA1* presented enhanced resistance to aphids due to increased levels of indole glucosinolates	Indole glucosinolates	[148]
*Brassica rapa*	*BrGI*	A *GI*-knockout mutant showed altered transcripts of glucosinolates as well as reduced accumulation	Aliphatic glucosinolates	[149]
*Arabidopsis*	*GI*	A *GI* mutant showed downregulation of *PAD4*	Camalexin	[154]

*AACT2*, *acetoacetyl-CoA thiolase*; ABA, abscisic acid; *ANS*, *anthocyanidin synthase*; *CCA1*, *CIRCADIAN CLOCK ASSOCIATED 1*; *DXS*, *1-deoxy-D-xylulose 5-phosphate (DXP) synthase*; *FT*, *FLOWERING LOCUS T*; *GI*, *GIGANTEA*; *HDR*, *1-hydroxy-2-methyl-2-butenyl 4-diphosphate (HMBPP) reductase*; *ISPS*, *isoprene synthase*; *LDOX*, *leucoanthocyanidin dioxygenase*; *LHY*, *LATE ELONGATED HYPOCOTYL*; *LNK*, *NIGHT LIGHT-INDUCIBLE AND CLOCK-REGULATED*; MEP, 2-*C*-methyl-D-erythriol-4-phosphate; *FLS*, *flavonol synthase*; *MYBD*, *MYB-like Domain*; MVA, mevalonic acid; *RVE*, *REVEILLE*; *PAD4*, *phytoalexin deficient 4*; *PRR*, *PSEUDO-RESPONSE REGULATOR*; *LCL5*, *LHY-CCA1-LIKE5*; *TOC1*, *Timing of CAB expression 1*.

## Data Availability

Data sharing is not applicable. No new data were generated or analyzed in this study.

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
