# Peer review of "Unlocking Nature’s Rhythms: Insights into Secondary Metabolite Modulation by the Circadian Clock"

_ijms, 2024, doi:10.3390/ijms25137308_

Round 1
Reviewer 1 Report
Comments and Suggestions for Authors
Dear Authors,
I have thoroughly reviewed your manuscript titled "Unlocking Nature’s Rhythms: Insights into Secondary Metabolites Modulation by the Circadian Clock." The manuscript is well-structured and provides valuable insights into the modulation of secondary metabolites by the circadian clock. However, there are a few areas that require minor revisions and clarifications to enhance the overall quality and readability of the manuscript. Below are specific suggestions and comments for improvement:
- Line 14: Replace "optically" with "optimally."
- Line 16: "analyzes" should be corrected to "analyses."
- Line 27: Consider replacing "circadian or internal clock" with "circadian clock."
- Consider rephrasing the last sentence for better readability, e.g., "This review explores the molecular mechanisms underlying the circadian rhythms of secondary metabolites, such as phenolic compounds, terpenoids, and nitrogen-containing compounds."
- Line 42: Use consistent abbreviations, i.e., "phyA/phyB" or "phyA and phyB."
- Line 43: "act" should be "acts."
- Line 49: "genomic levels that form" could be rephrased for clarity.
- Line 86: Add clarity on how PRR7 and PRR5 interact within the output pathway.
- Line 125: "multifunctional roles" could be specified with examples.
- Line 128: Include references for the detrimental effects on plant growth.
- Line 130: "have no fundamental role" might be better phrased as "are not essential for."
- Line 133: Consider simplifying the description of the interactive metabolic network.
- Line 141: Specify which environmental conditions affect the quantity and quality of secondary metabolites.
- Line 148: "2-C-methylerythriol-4-phosphate" should be corrected to "2-C-methyl-D-erythritol 4-phosphate."
- Line 454: "follow daily rhythms" could be more specific about which metabolites.
- Line 457: Clarify how RVE8/LCL5 and CCA1 interaction differs in terms of anthocyanin biosynthesis.
- Line 467: Consider highlighting key implications for climate change in crop breeding.
-
Could you elaborate on how the plasticity of the circadian oscillator impacts plant adaptability?
-
Are there specific stress conditions where PRR7 and PRR5 play a more significant role?
- Are there any specific non-model species that you believe should be prioritized for future research?
- How do you envision the practical applications of your findings in agricultural practices?
After addressing these comments and suggestions, the manuscript would be suitable for acceptance with minor revisions.
Comments on the Quality of English Language
The manuscript would benefit from a thorough review by a native English speaker or a professional editing service to correct grammatical errors and improve readability. Some sentences are complex and could be simplified for better understanding.
Author Response
Thank you very much for taking the time to review this manuscript and for the positive comments on our work and for the critical evaluation of our manuscript. We believe that the comments helped us in improving our manuscript. Please find the detailed responses below and the corresponding revisions/corrections highlighted in the re-submitted files.
Comment 1. Line 14: Replace "optically" with "optimally."
Response: We checked the entire document but could not find the word “optically” in the manuscript. In line 11 we use “optimally”.
Comment 2. Line 16: "analyzes" should be corrected to "analyses."
Response: The manuscript is written in American English.
Comment 3. Line 27: Consider replacing "circadian or internal clock" with "circadian clock."
Response: Has been modified as suggested.
Comment 4. Consider rephrasing the last sentence for better readability, e.g., "This review explores the molecular mechanisms underlying the circadian rhythms of secondary metabolites, such as phenolic compounds, terpenoids, and nitrogen-containing compounds."
Response: We have modified the sentence as suggested.
Comment 5. Line 42: Use consistent abbreviations, i.e., "phyA/phyB" or "phyA and phyB."
Response: Has been modified as suggested.
Comment 6. Line 43: "act" should be "acts."
Response: Here we mean phytochromes in general and not a specific phytochrome. To make this clearer, we have changed the sentence as follows: “In particular, phytochromes act additively in red light input to the clock while cryptochromes act redundantly in blue light input”(Line 43).
Comment 7. Line 49: "genomic levels that form" could be rephrased for clarity.
Response: We have modified the text to clarify the sentence: “The central oscillator is composed of ”canonical clock genes“ that work at various genomic levels that compose the central or core loop, the morning loop and the evening loop” (line 48).
Comment 8. Line 86: Add clarity on how PRR7 and PRR5 interact within the output pathway.
Response: The text has been modified including more information on the role of PRRs within the output pathway. Figure 1 has been modified accordingly too. The text has been modified as follows: “In fact, PRR9 and PRR7 along with PRR5 are part of the output pathway of the circadian clock, which involves transcription–translation feedback loops that regulate biological processes from seed germination to environmental stress responses (Figure 1). Particularly, PRR9 and PRR7 were found to be critical components of a temperature-sensitive circadian system [36]. A few years later, these findings were confirmed, where a triple mutant with loss of PRR9, PRR7 and PRR5 presented a higher expression of the dehydration-responsive element-binding protein 1 (DREB1) / C-repeat binding factors (CBFs), suggesting that these PRRs regulate cold stress responses by inhibiting DREB1/CBFs [37]. Moreover, both PRR5 and PRR7 have been recently reported to positively regulate Abscisic acid (ABA) signaling through the association with the transcription factor ABSCISIC ACID-INSENSITIVE5 (ABI5) during seed germination [38]. CCA1 and LHY also seem regulate cold stress responses, but instead of being repressors, they induce the expression of CBFs [39,40].”
Comment 9. Line 125: "multifunctional roles" could be specified with examples.
Response: We have included some examples as suggested: “Secondary metabolites are highly reactive bioactive compounds that play multifunctional roles in both defense and interactions of plants with the environment, such as antioxidants, osmoregulators, allelopathic effectors, antifeeding agents, etc.” (line 126).
Comment 10. Line 128: Include references for the detrimental effects on plant growth.
Response: We have included the following refence; Erb, M.; Kliebenstein, D.J. Plant Secondary Metabolites as Defenses, Regulators, and Primary Metabolites: The Blurred Functional Trichotomy. Plant Physiol 2020, 184, 39–52, doi:10.1104/pp.20.00433.
Comment 11. Line 130: "have no fundamental role" might be better phrased as "are not essential for”
Response: We have modified the text to clarify the sentence: “Although secondary metabolites are often referred to as organic compounds with no essential role in maintaining plant life processes, new studies have shown that the boundaries between primary and sec-ondary metabolism are blurring” (line 131).
Comment 12. Line 133: Consider simplifying the description of the interactive metabolic network.
Response: The aim of this manuscript is not to elucidate the interactive metabolic network between primary and secondary metabolism with hormone regulation. We have included “as reviewed in” to alert the reader that more details can be found in the review.
Comment 13. Line 141: Specify which environmental conditions affect the quantity and quality of secondary metabolites.
Response: We have included some examples as suggested: “Plants synthesize more than 100,000 secondary metabolites through different metabolic pathways, with quantity and quality varying depending on environmental conditions, including pathogen infections, herbivore attacks, cold stress, salinity, heat and drought stress” (line 142).
Comment 14. Line 148: "2-C-methylerythriol-4-phosphate" should be corrected to "2-C-methyl-D-erythritol 4-phosphate."
Response: Has been modified as suggested.
Comment 15. Line 454: "follow daily rhythms" could be more specific about which metabolites.
Response: Has been modified as suggested and we included examples: “Numerous secondary metabolites have been found to follow daily rhythms, such as carotenoids, flavonoids, and glucosinolates, with some of them being directly regulated by canonical clock genes such as CCA1 and TOC1 (Table 1).”
Comment 16. Line 457: Clarify how RVE8/LCL5 and CCA1 interaction differs in terms of anthocyanin biosynthesis.
Response: This does not represent an interaction between RVE8/LCL5 and CCA1 in anthocyanin biosynthesis, but RVE8/LCL5 activates the biosynthesis genes in a similar circadian rhythmic expression as CCA1.We have modified this part to clarify the idea: “Interestingly, the MYB-related transcription factor RVE8/LCL5 activates the anthocyanin biosynthesis genes with a similar rhythmic expression as CCA, showing a morning acrophase. Unlike CCA1, RVE8/LCL5 activates TOC1 expression, while LNK acts as its antagonist, inhibiting the anthocyanin biosynthesis genes.” (line 493-497).
Comment 17. Line 467: Consider highlighting key implications for climate change in crop breeding.
Response: A paragraph has been added in lines 503-517: “However, most molecular studies are still based on the model plant Arabidopsis, while in non-model species, direct evidence for the regulation of these secondary metabolites by circadian clock genes is lacking. This highlights the need for further genomic and transcriptomic studies in non-model species with cropping or ecolog-ical value to better understand these regulatory mechanisms and to implement local and targeted agricultural practices. Additionally, further research is needed on identifying secondary metabolites that alter the circadian period and/or phases. Understanding the intricate network of circadian regulation is essential for im-proving plant resilience and adaptation. In the context of climate change, this knowledge is particularly crucial for crop species. It can lead to more directed crop breeding and more efficient, cost-effective management practices that integrate circadian agriculture as a fundamental aspect. By incorporating these insights into breeding programs and agricultural management decisions, we can contribute to sustainable agricultural practices and enhance food security.”
Comment 18. Could you elaborate on how the plasticity of the circadian oscillator impacts plant adaptability?
Response: A reference on the plasticity of the circadian oscillator has been made in lines 83-88: “Plants that have functional oscillators but that cannot entrain well to the environ-ment, will struggle to adapt, leading to poor performance. The circadian oscillator is dynamically plastic, and it adjusts phase and period to abiotic and biotic signals [34]. Not only environmental cues alter the circadian period and phases but also some other metabolites such as sugars, hormones or ions can adjust the circadian oscillator.”
Comment 19. Are there specific stress conditions where PRR7 and PRR5 play a more significant role?
Response: These specific stress conditions have already been addressed in Comment 8 lines 86-97.
Comment 20. Are there any specific non-model species that you believe should be prioritized for future research?
Response: Non-model species with cropping and/or ecological value. Since we are advocating for a more sustainable agriculture that includes sustainable management practices, which include the culture of local species and cultivars, all local crops should be prioritized for future research. We have modified the text in the conclusions as follows “However, most molecular studies are still based on the model plant Arabidopsis, while in non-model species, direct evidence for the regulation of these secondary metabolites by circadian clock genes is lacking. This highlights the need for further genomic and transcriptomic studies in non-model species with cropping or ecological value to better understand these regulatory mechanisms and to implement local and targeted agricultural practices. Understanding the intricate network of circadian regulation is essential for improving plant resilience and adaptation. In the context of climate change, this knowledge is particularly crucial for crop species. It can lead to more directed crop breeding and more efficient, cost-effective management practices that integrate circadian agriculture as a fundamental aspect. By incorporating these insights into breeding programs and agricultural management decisions, we can contribute to sustainable agricultural practices and enhance food security.” Lines 491-510.
Comment 21. How do you envision the practical applications of your findings in agricultural practices?
Response: see response of comment 20.

Reviewer 2 Report
Comments and Suggestions for Authors
Comments to authors:
The authors should add graphic abstract.
The authors should have English editing to the manuscript.
Abstract:
- The abstract should be organized, i.e. give brief note to the introduction then highlight the problem, after that present the promising results and finally describe the future prospective.
- Please mention why this point of research is important and what is the impact to the environment and human kind?
Introduction
Lines from 33 to 41 please, add the reference next to its own sentence, the same in all the text.
Discussion
Discussion part is to be written to make it easier for the reader to understand the presented information of the secondary metabolites.
Conclusion
Clear future prospective should be added for further development in this area.
Comments on the Quality of English Language
minor English editing is recommended
Author Response
Thank you very much for taking the time to review this manuscript and for the positive comments on our work and for the critical evaluation of our manuscript. We believe that the comments helped us in improving our manuscript. Please find the detailed responses below and the corresponding revisions/corrections highlighted in the re-submitted files.
Comment 1. The authors should add graphic abstract.
Response: Has been added as suggested.
Comment 2. The authors should have English editing to the manuscript.
Response: The entire manuscript has been carefully edited in English.
Comment 3. Abstract: The abstract should be organized, i.e. give brief note to the introduction then highlight the problem, after that present the promising results and finally describe the future prospective. Please mention why this point of research is important and what is the impact to the environment and human kind?
Response: The abstract has been revised as suggested: “Plants, like many other living organisms, have an internal timekeeper, the circadian clock, which allows them to anticipate photoperiod rhythms and environmental stimuli to optimally adjust plant growth, development and fitness. These fine-tuned processes depend on the interaction between environmental signals and the internal interactive metabolic network regu-lated by the circadian clock. While primary metabolites have received more attention, the im-pact of the circadian clock on secondary metabolites remains less explored. Transcriptome analyses revealed that many genes involved in secondary metabolite biosynthesis exhibit diurnal expression patterns, potentially enhancing stress tolerance. Understanding the interaction mechanisms between the circadian clock and secondary metabolites, including plant defense mechanisms against stress, may facilitate the development of stress-resilient crops and enhance targeted management practices that integrate circadian agricultural strategies, par-ticularly in the face of climate change. In this review we will delve into the molecular mechanisms underlying circadian rhythms of phenolic compounds, terpenoids and N-containing compounds.” (line 11-23).
Comment 4. Lines from 33 to 41 please, add the reference next to its own sentence, the same in all the text.
Response: Has been modified as suggested.
Comment 5. Discussion part is to be written to make it easier for the reader to understand the presented information of the secondary metabolites.
Response: A discussion section is not necessarily common in a review article, unlike a research article. For our article, we do not see an improvement in the manuscript by including a discussion section and we think it would, in contrast, lead to many repetitions of what has already been mentioned. In addition, we are following the example of numerous other review articles in this journal that also do not have a discussion section, such as: https://doi.org/10.3390/ijms25136951; https://doi.org/10.3390/ijms25136963 (registering DOI); https://doi.org/10.3390/ijms25136993 (registering DOI).
Comment 5. Clear future prospective should be added for further development in this area.
Response: Has been modified as suggested: “. However, most molecular studies are still based on the model plant Arabidopsis, while in non-model species, direct evidence for the regulation of these secondary metabolites by circadian clock genes is lacking. This highlights the need for further genomic and transcriptomic studies in non-model species with cropping or ecological value to better understand these regulatory mechanisms and to implement local and targeted agricultural practices. Additionally, further research is needed on identifying secondary metabolites that alter the circadian period and/or phases. Understanding the intricate network of circadian regulation is essential for im-proving plant resilience and adaptation. In the context of climate change, this knowledge is particularly crucial for crop species. It can lead to more directed crop breeding and more efficient, cost-effective management practices that integrate circadian agriculture as a fundamental aspect. By incorporating these insights into breeding programs and agricultural management decisions, we can contribute to sustainable agricultural practices and enhance food security.” (line 503-517).
